# Measurement Report: Observations of long-lived volatile organic compounds from the 2019-2020 Australian wildfires during the COALA campaign

Asher P. Mouat[1], Clare Paton-Walsh[3], Jack B. Simmons[3], Jhonathan Ramirez-Gamboa[3], David W. T. Griffith[3] and Jennifer Kaiser[1,2]

[1]Department of Civil and Environmental Engineering, Georgia Institute of Technology, Atlanta GA 30332, USA

[2]Department of Earth and Atmospheric Sciences, Georgia Institute of Technology, Atlanta GA 30332, USA

[3]School of Earth, Atmospheric, and Life Sciences, University of Wollongong, Wollongong, NSW, Australia 2522

*Correspondence to:* Asher Mouat (amouat3@gatech.edu)

**Abstract.** In 2019/2020, Australia experienced its largest wildfire season on record. Smoke covered hundreds of square kilometers across the southeastern coast and reached the site of the COALA-2020 (**C**haracterizing **O**rganics and **A**erosol **L**oading over **A**ustralia) field campaign in New South Wales. Using a subset of nighttime observations made by a proton-transfer-reaction time-of-flight mass spectrometer (PTR-ToF-MS), we calculate emission ratios (ERs) and factors (EFs) for 15 volatile organic compounds (VOCs). We restrict our analysis to VOCs with sufficiently long lifetimes to be minimally impacted by oxidation over the ~8 h between when the smoke was emitted and when it arrived at the field site. We use oxidized VOC to VOC ratios to assess the total amount of radical oxidation: maleic anhydride/furan to assess OH oxidation, and (cis-2-butenediol + furanone)/furan to assess $NO_3$ oxidation. We examine time series of $O_3$ and $NO_2$ given their closely linked chemistry with wildfire plumes and observe their trends during the smoke event. Then we compare ERs calculated from the freshest portion of the plume to ERs calculated using the entire nighttime period. Finding good agreement between the two, we are able to extend our analysis to VOCs measured in more chemically aged portions of the plume. Our analysis provides ERs and EFs for 6 compounds not previously reported for temperate forests in Australia: acrolein (a compound with significant health impacts), methyl propanoate, methyl methacrylate, maleic anhydride, benzaldehyde, and creosol. We compare our results with two studies in similar Australian biomes, and two studies focused on US temperate forests. We find over half of our EFs are within a factor of 2.5 relative to those presented in Australian biome studies, with nearly all within a factor of 5, indicating reasonable agreement. For U.S.-focused studies, we find similar results with over half our EFs within a factor of 2.5, and nearly all within a 5, again indicating reasonably good agreement. This suggests that comprehensive field measurements of biomass burning VOC emissions in other regions may be applicable to Australian temperate forests. Finally, we quantify the magnitude attributable to the primary compounds contributing to OH reactivity from this plume, finding results comparable to several U.S-based wildfire and laboratory studies.

## 1 Introduction

Wildfire smoke significantly affects atmospheric composition, chemistry, human health, and radiative balance (Andreae and Merlet, 2001; Yokelson et al., 2008; Akagi et al., 2011; Ford et al., 2018; Gregory et al., 2018; Sokolik et al., 2019; Macsween et al., 2020). Wildfire season duration and intensity are predicted to increase in the future, suggesting a growing influence of biomass burning in coming decades (Fairman et al., 2015; Donovan et al., 2017; Abatzoglou et al., 2019). Volatile organic compounds (VOCs) emitted from biomass burning (BBVOCs) are directly harmful to human health and can contribute to the formation of

ozone and secondary organic aerosol (SOA) (Akagi et al., 2012; Keywood et al., 2013; Lawson et al., 2015; Sekimoto et al., 2017). Predictions of BBVOC emissions are complicated by the complexity of combustion and fuel characteristics, and model parametrizations are based on a limited number of field observations (Hatch et al., 2015; Sekimoto et al., 2018).

Australia wildfires emit 7-8% of global biomass burning emissions, producing more volatilized carbon than the United States and Europe, with smoke plumes significantly influencing local and even global air quality (Ito and Penner, 2004; Van Der Werf et al., 2010; Keywood et al., 2013; Lawson et al., 2015). In 2019/2020, Australia experienced its worst wildfire season on record with an estimated 19 million hectares of land destroyed (Filkov et al., 2020). This particular season is now colloquially known as the Black Summer, due to its prolonged intensity and length (Oct 2019 – Feb 2020). Many of Australia's major cities were blanketed in
smoke for weeks at a time, leading to long-term exposure to excessive concentrations of harmful atmospheric compounds (Borchers Arriagada et al., 2020). These fires predominantly affected the temperate forests of the state of New South Wales (NSW), burning the largest land area of anywhere in the country (Davey and Sarre, 2020). Despite the impact on atmospheric composition from Australian fires, its biomes remain understudied, particularly these same NSW forests (Lawson et al., 2015). Given the complexity and variability in biomass burning scenarios and the use of emission factors (EFs, in units of kg VOC emitted/kg fuel burnt) to
inform air quality models, this can lead to issues in effectively constraining emissions. For example, Lawson et al. (2017) reported a strongly non-linear response in simulated ozone ($O_3$) when varying biomass burning (BB) EFs, showing the resulting sensitivity from chemical transport models (CTMs). This sparseness of measurements leads to the use of North American EFs (such as those from Burling et al. (2011) or Akagi et al. (2011)) to inform CTMs, simulating emissions of geographically separate biomes. Even among similar biomes (for instance, the temperate forests of the U.S.), fuel types differ and thus can influence the speciation of
VOCs emitted (Coggon et al., 2016; Hatch et al., 2017; Guérette et al., 2018). Further evidence of this is found in a study by (Guérette et al., 2018) showing that EFs of some VOCs (e.g. formic acid, ethane, monoterpenes, acetonitrile) can be 3 – 5 times higher than those measured in the US, and attributing this to fuel type.

A complicating factor in deriving EFs from field observations is accounting for the influence of chemical processing. EFs are ideally based on observations close to the fire. When this is not possible, indicators of plume chemical age, such as oxidized VOC
(OVOC) to VOC ratios, can be used to diagnose the relative age of a plume. During the day, downwind VOC concentrations are primarily influenced by OH-initiated oxidation. At night, $NO_3$-inhated oxidation can significantly influence observed VOC concentrations (Decker et al., 2019; Kodros et al., 2020). There are several methods in existence for assessing daytime oxidation, but fewer are known for the night (De Gouw et al., 2006; Liu et al., 2016; Gregory et al., 2018; Decker et al., 2019). In this work, we use the maleic anhydride-to-furan ratio introduced in Gkatzelis et al. (2020) to assess OH oxidation. We examine the use of a
new OVOC/VOC ratio, cis-2-butenediol+furanone-to-furan, as an indicator of nighttime oxidation.

To further assess the effects of nighttime transport on biomass burning emissions, we look at the magnitude of OH reactivity measured that results from the compounds which most substantially contribute to it and determine the relative contributions of the resulting chemical groups. Certain categories of BBVOCs like furans or phenols, which are emitted in the combustion process, are important as they enhance OH reactivity and resultingly have high $O_3$ and SOA forming potential, and are considered to be
understudied (Gilman et al., 2015; Hatch et al., 2017). Most wildfire studies are conducted during the daytime, with plume oxidation focused on interactions with the OH radical and $O_3$ (Liu et al., 2016; Coggon et al., 2019; Palm et al., 2020; Decker et al., 2021; Permar et al., 2021). However, the plume studied here spent a significant amount of time transported under nighttime conditions.

Additionally, we use time series to observe chemical trends in ozone ($O_3$) and nitrogen dioxide ($NO_2$) as their emissions and chemical behavior are intimately linked with biomass burning chemistry. $O_3$ production in wildfire plumes is contingent on initial emissions, local environment, and atmospheric processing during transportation. Wildfire plumes emit significant precursors of $O_3$, but there is not a general consensus towards generation or depletion, with various campaigns reporting measurements in either case, especially in the instance of processed, downwind plumes (Verma et al., 2009; Alvarado et al., 2010; Jaffe and Wigder, 2012; Lawson et al., 2015; Brey and Fischer, 2016; Müller et al., 2016). $NO_2$ is emitted during the combustion process and has a non-linear relationship to $O_3$ production via reactions with these VOC precursors. However, the $NO_2$ radical has additional chemical pathways with OH, $NO_3$, and phenolic compounds leading to a general $NO_x$-limited regime (Jaffe and Wigder, 2012; Liang et al., 2022; Robinson et al., 2021). Furthermore, there are again fewer observations for the effect of nighttime oxidation processes on $O_3$ production with a recent modeling study conducted by Decker et al. (2019). $O_3$ production in wildfire smoke remains a significant source of uncertainty in its contribution to the tropospheric $O_3$ budget (Jaffe and Wigder, 2012; Young et al., 2018; Xu et al., 2021).

Here, we use observations from a proton-transfer-reaction time-of-flight mass spectrometer (PTR-ToF-MS) during the 2019-2020 Australian wildfire season to derive EFs of 15 compounds, including 6 compounds for which there are no previous observations. We examine a subset of smoke-influenced nighttime observations made by a PTR-ToF-MS during the COALA-2020 field campaign. $NO_3$-initiated oxidation dominated the chemical processing late in the night, as the plume travelled ~8 h to the field site from large, highly active fires to the south. We also use co-located FTIR measurements of $CO_2$, CO and $CH_4$ to derive these EFs for nighttime longer-lived VOCs ($\tau_{BBVOC+NO3} \geq$ average transport time). We compare these results with five related studies, two focused on Australian temperate forests, two focused on US temperature forests, and one reporting EFs used to represent temperate biomes across the globe. We find generally good agreement across several of these studies and discuss potential reasons for discrepancies seen in EFs for selected compounds.

## 2 Field Site and Instrument Description:

### 2.1 Field Site and Active Fires

The COALA-2020 field site was located in Cataract Scout Park (34.247° S, 150.825° E) at 400 m above sea level, 15 km inland, and 30 km to the northwest of the nearest urban area (Wollongong, NSW). Fig. 1 shows the field site relative to the fires active between 1 Feb and 5 Feb 2020. We use the Suomi VIIRS thermal anomalies product filtering for points at high confidence levels to avoid counting any reflective false positives from plains or urban centers. Also plotted is the normalized difference vegetative index (NDVI) which is determined from measurements aboard the MODIS Terra satellite (Didan, 2021). The fires are primarily located in temperate forests along the southeastern coast, with a small inland group near Canberra. These forests consist of open, tall woodlands made up of *Eucalyptus* species grouped generally as dry sclerophyll.

### 2.2. PTR-ToF-MS and supporting observations

VOCs were measured using an Ionicon PTR-ToF-MS 4000 which operated with a mass resolution between 2000-3000 FWHM m/$\Delta$m and at a mass range spanning m/z = 18-256. The drift tube was held at a temperature of 70° C, pressure at 2.60 mbar, and

an $E/N$ = 120 Td (electric field to molecular number density ratio). The instrument was housed in a climate-controlled unit, connected to a 15 m long, ¼" OD PTFE insulated line attached to a 10 m tall mast, placing inlet height 0.5 m above canopy height Sample flowed through the inlet at 3 SLPM for a residence time of 2.5 s. Peak separation of 1 min averaged spectra was conducted in Ionicon's PTR-Viewer 3 software.

Calibrations were performed using two VOC cylinders designed by Airgas on 31 Jan 2020, three days before measuring the smoke event discussed here. A second calibration was performed in the following week with little change in instrument sensitivity. The cylinders contained 17 compounds spanning a mass range of 33-154 Da and are shown in Table S1. Many of these compounds are reported in the final EFs list – methanol, acetonitrile, acetaldehyde, acrolein, acetone, MVK+MACR, benzene, C8-aromatics, and C9-benzenes. All compounds used either do not fragment under these drift tube conditions or have known fragmentary peaks. Instrument zeros were determined using ultra-zero air. Limits of detection ($3\sigma$) for calibrated species are also given in Table S1 and range between 5-165 ppt. The raw counts per second (cps) were corrected for instrument transmission, which was determined using a subset of the species in the calibration standards. Corrected cps are then normalized to the reagent ion signal ($H_3O^+$ ccps x $10^6$, ncps) using the methodology described by Sekimoto et al. (2017). For compounds of interest not included in the calibration standards, we use the method described by Sekimoto et al. (2017), which yields uncertainties at 100%. Table S2 shows all compounds presented in this study alongside whether they were included in the calibration standards and their respective uncertainty.

In addition to the PTR-ToF-MS measurements, we use observations of CO, $CO_2$, and $CH_4$ obtained from the collocated FTIR system. Information of this instrument and its setup is provided in Griffith et al. (2012).

## 3. Observed CO, VOC, and OVOC Enhancements

Fig. 2 shows the observations of CO and VOCs during a smoky period on 3 – 4 Feb 2020. CO and acetonitrile – long-lived tracers associated with wildfires (Coggon et al., 2016) – are used to identify the total period of time during which observations were impacted by smoke. Enhancements in both species start at 17:30 local time on 3 Feb and lasting until 19:00 on 4 Feb, when wind direction shifted.

We use furan, a short-lived smoke tracer, and its oxidation products to determine which periods of the smoke event represent the least oxidized plume. Furan is highly reactive with OH ($k_{OH + furan}$ = at 4.04 x $10^{-11}$ $cm^3$ $molec^{-1}$ $s^{-1}$ 298 K and 1 atm) and $NO_3$ ($k_{NO3 + furan}$ = at 1.36 x $10^{-12}$ $cm^3$ $molec^{-1}$ $s^{-1}$ at 298 K and 1 atm). OH-initiated oxidation produces maleic anhydride, which has low reactivity with both OH and $NO_3$ ($\tau_{OH}$ = 3.99 days, $\tau_{NO3}$ = 1.42 days with $[OH]_{Avg}$ = 2 x $10^6$ molec $cm^{-3}$ and $[NO_3]_{Avg}$ = 8 x $10^7$ molec $cm^{-3}$, with reaction rate constants from Grosjean (1992) and Bierbach (1994) and no reported direct emissions). The ratio of maleic anhydride-to-furan therefore provides a relative measure of the plume photochemical age. Using aircraft-based observations of wildfire plumes in the Western US, Gkatzelis et al. (2020) found that maleic anhydride-to-furan ratios below 0.10 indicate the plume has undergone little OH processing.

Nighttime in-plume furan oxidation is dominated by $NO_3$, with contributions from $O_3$ (Decker et al., 2019). While many BBVOCs are highly reactive with $NO_3$, there is substantially less research on indicators of $NO_3$ oxidation. Decker et al. (2019) track $NO_3$ chemistry using the ratio of total reactive nitrogen ($NO_y$) to $NO_x$, and Kodros et al. (2020) examine $NO_3$-reacted products such as nitrocatechol and nitrophenol of phenolic compounds (e.g. phenol, catechol, cresol). Measurements of $NO_y$ were not made during

this field campaign, and $NO_3$-products of phenols were subject to high uncertainty due to fragmentation in our PTR-ToF-MS measurement. We therefore examine a new indicator of $NO_3$ processing using furan's dominant $NO_3$ products – cis-2-butenediol and furanone (Berndt et al., 1997). Both products are relatively long lived, with lifetimes estimated at $\tau_{cis-2-butenediol}$=9 days and $\tau_{furanone}$= 8 h assuming an average concentration of $[NO_3] = 8 \times 10^7$ molec $cm^{-3}$ (O'dell et al., 2020). Lab based studies and field campaigns conducted in the US and Australia suggest that furan and furanone EFs are comparable, with study-averaged values for furan ranging from $0.132 – 0.51$ g $kg^{-1}$ and $0.27 – 0.57$ for furanone (Andreae and Merlet, 2001; Akagi et al., 2011; Hatch et al., 2015; Stockwell et al., 2015; Liu et al., 2017; Koss et al., 2018; Selimovic et al., 2018). No furan EFs have been reported for Australian temperate forests and only one furanone EF is reported from Lawson et al. (2015) at a comparable value at $0.57$ g $kg^{-1}$. Additionally, emissions modeled in Decker et al. (2019) from wildfires suggest that furan and furanone are emitted in roughly equal proportions. As such, we operate not on the assumption of negligible OVOC emissions, but that variability in OVOC/VOC ratios are driven by chemical aging.

Fig. 2 shows furan enhancements, which begin later on Feb 3 than acetonitrile, maleic anhydride, and m/z 85 enhancements, indicating a less oxidized plume was being sampled. Maleic anhydride concentrations are high during the initial period of the smoke event, suggesting significant OH-initiated processing throughout the day before the plume reached the site. After sunrise, furan decays faster than CO, and maleic anhydride concentrations begin to rise, again showing the impact of OH-initiated oxidation. Cis-2-butenediol and furanone are both measured at m/z 85, and here on will be denoted as such. Enhancements in m/z 85 are seen when the smoke arrives and vary throughout the night. Just prior to sunrise (04:00 – 06:15, local time), both OVOC/VOC ratios rapidly decrease (Fig. 3), corresponding with a rise in furan, CO, and acetonitrile. Maleic anhydride/furan drops to 0.05, which is within the lower range of the chemically younger plumes reported by Gkatzelis et al. (2020). The ratio of m/z 85 to furan is around 2.5. While we cannot use this to quantify plume age since the two products are measured as a sum, we note that this period constitutes the lowest ratio throughout the event, with surrounding periods having ratios $1.6 – 2.8$ times greater. We note that at a value of 2.5, this plume has likely undergone significant aging, despite this being the freshest smoke detected during the campaign. Further corroboration of these results, determined via particulate matter (PM) composition, can be found in Simmons et al. (2022) (submitted). In their study, a ToF-ACSM was employed and observed the ratio of $PM_1$ mass fraction at mass-to-charge ratio 44 ($f_{44}$), where a lower mass fraction indicates a less oxidized plume. A similar decrease at $f_{44}$ in the same timeframe as the m/z 85 and maleic anhydride tracers is noted.

The rapid decreases in OVOC/VOC ratios are unlikely to result from shifts in chemistry alone. Instead, this suggests a shift in meteorological conditions which brings in smoke from a closer source, in agreement with measured wind direction, which shifted from flowing northeast to north at this time. We further investigate plume transport using a back-trajectory model.

## 4. Plume Origin and Transport Time

We use a HYSPLIT back-trajectory model (Stein, 2015) to determine the origin and transport time of the smoke arriving at the site throughout the smoke event. The meteorological input used is the Global Data Assimilation (GDAS) dataset. The model was set to assess trajectories at three different altitudes at 10 m, 500 m, and 1500 m above ground level (agl) to capture plume height. Our period of interest spans from 17:00 Feb 3, just before CO enhancements are seen at the site, to 06:00 4 Feb when furan concentrations rapidly decrease. The model was set to calculate a new 12 h trajectory every hour during this time. Back trajectories

are shown in Fig. 1. For every hour in the event (each represented by a color), one can track the origin of the sampled airmass 12 h in advance of its arrival.

A shift in trajectories occurred between 17:00 and 18:00 3 Feb, corresponding with the arrival of the smoke plume as indicated by observed CO enhancements. Subsequent trajectories originate near the fires located ~230-375 km from the field site on the southeast coast. The model shows that air masses initially kept at low altitude and were lofted to ~560 m agl when passing over the active fires ~25 km to the south, near Canberra (Fig. S1). The plume descended to 10 m agl as it reached the coast. The model suggests smoke sampled later in the evening (between 04:00 – 06:00 4 Feb) spent more time over land compared to previous points in the event. This shift in trajectories and the increasing intensity of fires near Canberra during this time means possible contributions to the decrease in OVOC to VOC age marker ratios. Further investigation is conducted via HYSPLIT forward trajectories in the supplement (Figures S2 and S3). In short, during this period, plumes from the Canberra fires were lofted to 2000 m agl well before crossing with the SE-fires plume, which attained a maximum altitude up to 560 m agl. This indicates little influence from the Canberra fires on our measurements. Given that there are two major clusters approximately 70 km apart in the SE, the influence of precipitation and wind speed (Figures S4 – 6) is considered to determine whether combustion conditions were comparable. Both fires experienced similar total precipitation in the month prior and experienced similar wind speeds during this smoke plume event. As a result, we conclude that combustion conditions are similar and that EFs derived from this plume would be representative of a biome average. Over the entire course of the event, HYSPLIT analysis suggests transport time from the fires to the field site is around 8 h (>200 km), but potentially shorter for the time frame immediately prior to sunrise.

## 5. $O_3$ and $NO_2$ Time Series

Detailed time series of $O_3$ and $NO_2$ are presented in this section in Fig. 4. Information regarding instrumentation and corresponding setups can be found in Section 2.1 of Simmons et al. (2022) (submitted). Like Fig. 2, a CO time series is provided to outline the general trend of smoke during the event.

A non-smoke influenced daytime and nighttime average (composed of 8 hr averages) was calculated for $O_3$ and $NO_2$ concentrations using data from the month of March. Smoke around the continent had been either transported or removed by rain by this time. $O_3$ was calculated to have a daytime concentration of 24.6 ppb and a nighttime concentration of 19.5 ppb. Respective concentrations were calculated for $NO_2$ at 2.2 ppb in the day and 3.3 ppb in the evening. Additionally, averages for a larger suite of gas and aerosol phase variables over all smoke events sampled during the COALA-2020 campaign can be found in Simmons et al. (2022) (submitted).

As stated before, smoke-related enhancements are visible around 17:30 local time in Fig. 4(d), with the hours prior being virtually devoid of tracers. Enhancements pick up without a shift in wind direction, with winds at this time travelling to the northwest, consistent with the HYSPLIT trajectories presented in Fig. 2. As the wind approaches a more easterly direction, enhancements in CO are maintained and concentrations of more reactive BBVOCs begin to increase. $O_3$ concentration on 3 Feb reaches a maximum of approximately 25 ppb around 14:00 local time and maintains this level until sunset where it decreases as biogenic sources are no longer emitting and photolysis is halted. $O_3$ concentration reduces to a minimum 15.6 ppb and $NO_2$ decreases down to 0.8 ppb both around midnight and both below the nighttime monthly average despite enhancements in CO. $O_3$ has a $R^2 = 0.48$ with CO and, when considering the known transport time of this smoke, indicates transportation rather than local production. Given the

comparatively low concentrations of both compounds at this time, it is likely that this plume is depleting these species. This is compounded with the low concentrations of $NO_2$ in this temperate forest setting and, despite emitting $NO_x$, wildfire plumes being generally $NO_x$-limited (Jaffe and Wigder, 2012; Robinson et al., 2021).

Around 03:30, the wind shifts from traveling northwest to west, significantly enhancing $O_3$, $NO_2$, CO, and total VOC concentrations, corresponding to the least aged portion discussed in Section 3. Sunrise occurs around 06:30 coinciding with a steady decline in highly reactive VOC enhancements (Fig. 2) and $NO_2$ (Fig. 4(c)). Liang et al. (2022) found a significant correlation of $R^2$=0.86 between $NO_2$ and maleic anhydride for a transported plume of similar age oxidized in the daytime. The opposite trend is observed in our scenario despite our measurements exhibiting comparable trends from maleic anhydride. The $NO_x$-limited environment and differences in biogenic VOC (BVOC) quantities arising from the forest setting in this study and the urban setting in theirs are likely responsible for the opposing trends in the $NO_2$ time series. Maleic anhydride similarly peaks around noon on the 4[th], and both its production and the fast depletion of furan indicate that OH chemical pathways generally oxidize this plume faster than $NO_3$ reaction pathways. While $O_3$ concentration continues to increase after sunrise, it cannot be stated that this is dominantly due to BBVOC oxidation given the strong source of BVOC emissions from the surrounding forest. Isoprene nitrates sequester $NO_2$, ultimately leading to $O_3$ production. The diel cycle of $O_3$ and isoprene on a non-smoke affected day strongly correlate to temperature and photoactive radiation. $O_3$ does achieve a max concentration of 30 ppb at 12:00 4 Feb, which is approximately 5.5 ppb above the daytime average and higher than the prior day despite similar temperatures (23.6 C on the 3[rd], and 24.5 C on the 4[th]). This most likely results from the combination of transported $O_3$ compounded with enhanced reactivity from the plume plus local, biogenic-related production. The plume is diluted at a consistent rate until 18:00 4 Feb when a shift in wind direction significantly reduces CO enhancements and concludes the smoke event.

## 6. Emission Factors

### 6.1 Species selection

To identify compounds which would be suitable for EF derivation, we compare the list of measured ions with compounds identified in previous literature such as Brilli et al. (2014), Hatch et al. (2015), Gilman et al. (2015), Stockwell et al. (2015), Bruns et al. (2017), Koss et al. (2018), and the PTR Library (Pagonis et al., 2019). To corroborate species assignment, we examine correlations of identifiable compounds with CO, acetonitrile, furans, and phenolic compounds which are well-established smoke tracers. We also examine tracer-tracer relationships, for instance the anti-trend between maleic anhydride and furan resulting from OH oxidation. We exclude compounds with low proton affinities that are known to have humidity-dependent calibration factors (e.g., HCHO, HCN). This results in 150 identified VOCs species measured and identified during the smoke event.

We further filter our VOC list by two criteria. First, VOC + $NO_3$ reaction rates must be included either in the NIST Chemical Kinetics Database (Manion, 2015) or Master Chemical Mechanism (v3.3.1) (Bloss et al., 2005; Jenkin, 1997; Jenkin et al., 2003; Saunders et al., 2003). Second, the VOC must have a significantly long lifetime against $NO_3$ oxidation to be minimally impacted over the 8 h transit time from the active fires to the field site ($\tau_{BBVOC+NO3}$<8 h, again assuming [$NO_3$] = 8 x $10^7$ molec cm$^{-3}$). This limits subsequent analysis to 15 nighttime long-lived VOCs.

### 6.2 Calculating Emission Ratios

An ER is defined here as the slope of a linear regression of a given VOC to CO (both in units of ppb). Following Guérette et al. (2018), ERs are reported if correlation between a given VOC and CO are well correlated, with $R^2 \geq 0.5$. High correlation minimizes the impact of the choice of regression method (e.g. orthogonal, York) on calculated slopes (Wu and Yu, 2018), and removes the need to account for background corrections (additional discussion of surrounding influential sources can be found in Simmons et al. (2022) (submitted)). We use a reduced major axis regression to determine emission ratios. Given the time component that affects our measurements, it should be noted that compounds with low emission factors and high reactivity are likely to be excluded as they have been reacted away before reaching the site, thus exhibiting an insufficient CO correlation.

We first derive ERs using all data from the "freshest" portion of the plume as determined from OVOC/VOC ratios (Marked "D" in Fig. 2). This produces 15 ERs that meet our criteria. We expect this period to provide the most accurate representation of original VOC emissions. We then calculate ERs for more aged portions of the smoke event (Periods A-C, Fig. 2), performing regression analysis on the chemically distinct time periods. The start and end time of each period is determined by visual inspection of VOC/CO behaviors, which all exhibit similar distinct periods. Fig. 5 provides an example of the analysis using acrolein. We average the slopes from each of these lines to derive an average ER for the full smoke event and compare to just the freshest portion of the plume (Period D). We find that using only the freshest smoke compared to using all the data generates very similar results for 9 of the 15 compounds (of which these 9 all have multiple ERs over the evening). Relative differences of the resultant ERs are within 1.5 – 47 % with two outliers: C8-aromatics (88%) and C3-benzenes (212 %). Three compounds have only 1 ER from all 4 periods (maleic anhydride, benzaldehyde, and creosol) so there is no standard deviation, but the remaining compounds from period D are captured within 1σ of ERs from periods A-D (shown in Fig. S7). Good agreement between methods allows us to extend our analysis beyond the freshest part of the plume, and therefore allows us to report ERs for a larger number of compounds. When focusing only on the freshest part of the plume, maleic anhydride, and benzaldehyde must be excluded due to insufficient $R^2$ with CO. All ERs reported here and used in EF calculation use the "average over evening" method and include these compounds. Additionally, only one ER for $CO_2$ and $CH_4$ have been calculated using the dataset from periods A-D. Both these compounds are long-lived, and from visual inspection, they do not form distinct time periods like the VOC ERs (shown in Fig. 4). A table with the resultant VOC ERs is also provided in the supplement (Table S3). We use the $CO_2$ ER to determine an average modified combustion efficiency with the following equation:

$$MCE = \frac{ER_{CO2}}{ER_{CO2}+ER_{CO}} \qquad (1)$$

where the $ER_{CO}$ is just unity and $ER_{CO2/CO}$ is 10.82 ppb$CO_2$ ppb$CO^{-1}$. This results in an MCE calculation of 0.92, indicating a less efficient, even mixture of smoldering and flaming (Akagi et al., 2011).

### 6.3 Calculating Emission Factors

Emission factors are defined as the mass of some trace gas emitted per mass of dry biomass burnt. The most direct way of calculating this quantity is capturing total emissions released from a fire as well as knowing the quantity of fuel burnt. Unless experiments are conducted in a laboratory setting, these quantities are not known. As such, emission factors are calculated

according to the carbon mass balance method (Akagi et al., 2011; Selimovic et al., 2018), using CO as the reference gas for the 15 reported species which produces the following equation:

$$EF_X = F_{carbon} \times 1000 \times MM_X/MM_C \times ER_{X/CO} / \sum ER_{Y/CO} \qquad (2)$$

where $F_{carbon}=0.5$ and is the assumed carbon fractional content of the fuel as used in previous studies (Akagi et al., 2011; Paton-Walsh et al., 2014). $MM_X$ is the molar mass of compound X, $MM_C$ is the molar mass of carbon, $ER_{X/CO}$ is the CO ER of X, and $\sum ER_{Y/CO}$ is the sum of $ER_{CO2/CO}$, $ER_{CH4/CO}$, and $ER_{CO/CO}$. These ERs constitute the major volatilized carbon components of the plume, but the resulting EFs may be overestimated by 1-2% (Andreae and Merlet, 2001) as this method assumes all volatilized carbon is detected including particulate carbon, VOCs, CO, and $CO_2$.

EFs derived in this work are presented in Table 1 alongside results from 2 eastern Australia-based studies by Lawson et al. (2015) and Guérette et al. (2018), 2 western US-based studies sampling emissions from corresponding temperate fuel types by Liu et al. (2017) and Permar et al. (2021), and 1 study by Akagi et al. (2011) that provides EFs for general temperate zones. Additionally, Fig. S8 displays these results via scatter plot.

First, in comparison with the Australia-based studies, Guérette et al. (2018) reports EFs notably larger than those presented in this work, with only benzene and C8-aromatics showing good agreement. Except for these two compounds and C3-benzenes, Guérette et al. (2018) reports larger EFs than Lawson et al. (2015) and none within agreement. Our results more closely agree with Lawson et al (2015) with methanol, acetone, and furanone EFs within 1σ, and acetonitrile and acetaldehyde falling within a factor of 2. This agreement is likely due to both this work and Lawson et al. (2015) examining opportunistically intercepted smoke plumes that experienced some processing whereas Guérette et al. (2018) sampled near-source, controlled ground burns. Guérette et al. (2018) reports an acetonitrile EF ~4.5 times higher than this work and ~3 times greater than Lawson et al. (2015) constituting one of the largest disparities. This is attributed to the native and abundant Acacias which are N-fixing species located mainly in forest understories. Their measurements likely had a higher proportion of this foliage constituting the total fuel load due to both proximity to the forest floor and resulting leaf litter. Another of the largest differences is MVK+MACR, which shows a disparity of ~6 times this work and 3 times that of Lawson et al. (2015). This is also most likely explained by differences in sampling approach in that proportional contributions of vegetation vary and plumes in Guérette et al. (2018) did not undergo any dilution or photochemical processing.

In comparison with US-based studies, methanol, acetonitrile, acetone, and benzene agree across both studies within 1σ, with acrolein, methyl propanoate, methyl methacrylate, C3-benzenes, and creosol agreeing very well with values reported by Permar et al. (2021). It should be noted that though within the estimated uncertainties, the value for creosol reported by Permar et al. (2021) is ~3.5 times greater than the value in this work, which constitutes another of the largest disparities in this dataset. Additionally, methanol agrees well with the value from Akagi et al. (2011). The EF for m/z 85 in this work is also expectedly larger than both other values presented here at ~3 times greater than Permar et al. (2021). This is likely due to the plume sampled in this work undergoing the longest transport of any plumes measured in other studies.

Perhaps an unexpected finding is that EFs derived in this work agree better with observations in the US than the Guerette et al. (2018) study, which was in the same region as the COALA-2020 measurements. It should be noted that all studies except Guerette et al. (2018) are from plumes sampled several km downwind. Differences previously characterized as arising from varying fuel

types may actually result from measurement approaches to deriving EFs and proximity to emission source. Agreement across results from this work and from the US-based studies lends credence to the use of newly presented EFs for modeling purposes in temperate Australian forests. Further corroborating this notion is the extremely good agreement (all EFs within uncertainty for all three studies) found between EFs in this work and those presented in Stockwell et al. (2015) and Koss et al. (2018). These results can be seen in the supplement in Fig. S9.

## 7. OH Reactivity

As this plume has been shown to oxidize faster when exposed to the OH radical as opposed to the $NO_3$ radical, this indicates that the nighttime transport of this plume would be able to comparably preserve OH reactivity. We investigate this by first determining which compounds were most significant in their enhancements, and then determine their corresponding OH reactivity.

First, a subset of the PTR-ToF-MS data was created by calculating ERs using the methodology described in Section 6.2 over the same nighttime period. However, we did not filter out compounds by their atmospheric lifetime, and any unidentified species were not considered regardless of correlation strength. This means the resulting OH reactivity is likely to be slightly low, but this method ensures reactivity solely from compounds attributable to BB emissions is being gauged. Then, an average for each compound was calculated using the same period for ERs. These nighttime averages were then compared with their diurnal cycles calculated using data from 1 – 19 Mar 2020 (ending date of PTR-ToF-MS sampling ambient air). If a compound's mean over the smokey period is greater than the mean of its diel cycle plus $1\sigma$ over the same timeframe, this compound is considered in the transported OH reactivity. Finally, we background correct the nighttime concentrations using the March diurnal cycles and convert to reactivity using equation 3:

$$R_{OH} = \sum [VOC_i] * A * k_{VOCi+OH} \qquad (3)$$

where $[VOC_i]$ is the concentration of the $i$th VOC in units of parts-per-billion, A is the conversion factor to molecules$_i$ cm$^{-3}$ (A = 2.46 x $10^{10}$ in units of molecules$_{air}$ cm$^{-3}$ ppb$^{-1}$ at 1 atm and 25 C), and k$_{VOCi + OH}$ is the OH rate constant for the corresponding VOC$_i$. Rate constants were again sourced from the same databases as the $NO_3$ rate constants. The rate constant used for m/z 85 was determined as an average of the constant provided in Koss et al. (2018) (k$_{OH}$ = 44.2 x $10^{-12}$ cm$^3$ molec$^{-1}$ s$^{-1}$) and Bierbach et al. (1994) (k$_{OH}$ = 52.1 x $10^{-12}$ cm$^3$ molec$^{-1}$ s$^{-1}$) assuming both compounds contributed equally to signal at this mass peak.

Ultimately, 26 compounds were determined to have the most significant contributions, transporting an average OH-reactivity of 5.25 s$^{-1}$, with a minimum of 3.15 s$^{-1}$ occurring around 03:00 4 Feb and a maximum of 9.83 s$^{-1}$ around 20:00 3 Feb, shown in Fig. 6. These values are well within range of those seen in nighttime and aged, daytime transported plumes by Liang et al. (2022), who measured a total OH reactivity range from approximately 4 – 26 s$^{-1}$. We calculate an OH reactivity from the primary biogenic VOCs (isoprene plus monoterpenes) for further comparison. The maximum biogenic value, achieved around 12:00 4 Feb, is 6.35 s$^{-1}$ and the average biogenic reactivity over the course of the campaign is 5.90 s$^{-1}$, indicating that the nighttime conditions allowed for the transport of a reactivity quantity that approximately doubled OH reactivity at the COALA-2020 field site. Additionally, there is little variability in the relative contributions to reactivity across these different groups over the course of the smoke event, indicating the plume experienced a consistent oxidation over the course of its travel.

Compounds from the plume have been grouped into four categories to capture their diversity. Expectedly, biogenic emissions contribute the most to total reactivity (attributable dominantly to isoprene), but the furans group is the most reactive with values from 1.24-3.93 $s^{-1}$. This group contains various furans (furan, 2-methylfuran, m/z 85, and furfural alcohol) wherein m/z 85 is by far the most significant contributing up to 69% of the group total. This high m/z 85 presence explains why this group is also the most OH reactive as most furans are largely oxidized by $NO_3$ during this transport timeframe, except m/z 85 which has a long $\tau_{NO3}$ but a comparatively shorter $\tau_{OH}$. The furan reactivity range is comparable to lab-based values measured in Gilman et al. (2015) which ranged from 1.3 – 5.5 $s^{-1}$. Both these studies find lower furan reactivities than lab measurements made in Koss et al. (2018) at an average reactivity of 14.2 $s^{-1}$, where furans constitute the third highest reactivity group. Aromatics make up the second most reactive group (range of 0.66 – 2.14 $s^{-1}$) in this study, with dominant contributions from phenol (39%), styrene (33%), and catechol (32%). Catechol's contribution is likely less than this as other studies have revealed that it shares significant portion of its mass peak with 5-methyl furfural (Stockwell et al., 2015; Koss et al., 2018). Despite their high $NO_3$ reactivity, phenolic compounds still dominate the overall OH reactivity contributions in this category. These compounds appear across other studies as primary contributors to OH reactivity (Gilman et al., 2015; Hatch et al., 2017; Koss et al., 2018; Sekimoto et al., 2018; Decker et al., 2019; Liang et al., 2022). Alkenes (range of 0.86 – 1.83 $s^{-1}$) are on par with aromatics, for which their reactivity is largely attributable to propene and butene, followed last by non-aromatic oxygenates (range of 0.28 – 1.87 $s^{-1}$), which contain compounds like methanol, acetaldehyde, and acetic acid. The comparably low reactivity from this group is unexpected as other studies have shown that the dominant contributions to reactivity come from this group (Gilman et al., 2015; Koss et al., 2018; Liang et al., 2022).

## 8. Conclusions

EFs were derived for a total of 15 trace gas species via measurements from a PTR-ToF-MS and an FTIR spectrometer, the resulting OH reactivity of the transported plume quantified, and $O_3$ and $NO_2$ time series investigated. The COALA-2020 ground-based field campaign opportunistically sampled a sustained biomass burning plume from 3 – 4 Feb 2020 during the 2019-2020 wildfire season in New South Wales, Australia. We determined via HYSPLIT trajectories that the most likely pathway traveled by the plume was from a distance ranging from ~230-375 km south from fires along the temperate forests of the east coast with contributions from more inland fires near Canberra, Australia. This plume lofted to an altitude of 500 m agl as it passed over active fires ~8 h out from the field site, before descending to 10 m agl while traveling over the ocean and reaching the site at 17:30 local time. All data used in the derivation of EFs was limited from sunset on 3 Feb to sunrise on 4 Feb as this period showed the greatest enhancements of reactive BB tracers like furan. Through visual inspection, we partitioned this plume event into 4 portions, and calculated and averaged the individual ERs. We used two age marker ratios derived from furan radical oxidation to determine the freshest portion of the plume and found that ERs from this portion corresponded well with the averaged ERs (within 1σ). Using EFs from the entire evening allowed for the inclusion of three more VOC EFs into this analysis which, for the freshest portion of the plume, did not meet the selection criteria for ERs.

We have further characterized wildfire emissions in Australia's temperate region by providing a more comprehensive suite of biome-averaged VOC EFs. This suite introduces new EFs for acrolein, methyl propanoate, methyl methacrylate, maleic anhydride, benzaldehyde. and creosol. Of particular note is acrolein, which has been shown to be a gas-phase variable posing significant harm to human health (O'dell et al., 2020; Simmons, 2022). When compared with values reported from 2 Australian studies located in

the same or nearby temperate forests, we find mixed agreement with results from Guérette et al. (2018) as only two values are captured within our EF variability, with acetonitrile differing by a factor of ~4.5 times and MVK+MACR differing by ~6 times. However, 2 compounds are within the range of variability for Lawson et al. (2015) and 2 others are well within a factor of 2, which indicates reasonable agreement. Furthermore, comparison with two recent US studies that report data on analogous temperate zones, as well as one report covering global temperate regions, show generally good agreement for 9 of the 15 compounds, with several others within a factor of 2, indicating very good agreement. This closer agreement with these studies, as well as that of Lawson et al. (2015), is likely due to the measurement approach when deriving EFs as both US-based studies were aircraft campaigns, and the Australia-based study intercepted a transported plume much like this work. Guérette et al. (2018) sampled controlled burns on a ground campaign virtually at the emission source. This indicates that variability previously ascribed to differing fuel types may be overshadowed by sampling approach and that comprehensive measurements from US-based studies may be useful for studying Australian biomes. Agreement with both Lawson et al. (2015) and the US-based studies indicates that results here are valid for future use in Australian, biome-specific biomass burning studies. Compounding this is the excellent agreement found between EFs in this study and a comparison of two laboratory, U.S.-based, temperate fuel studies, indicating the potential for lab-based results to be similarly applicable. Chemically comprehensive near-source observations of Australian fuel types are needed to evaluate the importance delineating temperate forest EFs in different regions across the globe.

Probing the OH reactivity of the plume revealed that the nighttime conditions, despite the long transport time, transported a quantity that effectively doubled OH reactivity at the COALA-2020 field site, with contributions arising from expected classes of compounds such as furans (most contribution), aromatics (second), and alkenes (third). m/z 85 contributed most significantly of the furans measured, which is due to its long $NO_3$-lifetime but short OH-lifetime. Other furans had largely been reacted away before reaching the COALA-2020 field site. Phenol had the largest contribution of the measurable phenolic compounds despite its high $NO_3$ reactivity. Alkenes and aromatics were found, as a group, to have an on par reactivity and, unexpectedly, non-aromatic oxygenates contributing the least.

*Data Availability*. Data are available from PANGAEA archive at https://doi.pangaea.de/10.1594/PANGAEA.927277.

*Supplement*. The supplement related to this article is available online at:

*Author Contributions*. Asher P. Mouat conducted PTR-ToF-MS measurements and subsequent data analysis. Jack Simmons, Clare Paton-Walsh, and Jhonathan Ramirez-Gamboa oversaw the maintenance and in-person operation of the PTR-ToF-MS for much of the COALA-2020 field campaign. CO measurements were provided by David Griffith. Clare Paton-Walsh led the COALA-2020 campaign, whilst Jennifer Kaiser led PTR-ToF-MS instrument deployment and data analysis. All coauthors have provided substantial input during the process of drafting this work.

*Competing interests*. The authors declare that they have no conflict of interest.

*Acknowledgements*. This work was supported by NSF grant 2016646. We thank Travis Naylor, Ian Galbally and all the UOW COALA-2020 team for their aid in conducting measurements during the field campaign and all input thereafter. We additionally gratefully acknowledge the NOAA Air Resources Laboratory (ARL) for providing the HYSPLIT transport and dispersion model

used for analysis in this publication. We acknowledge the use of data and/or imagery from NASA's Land, Atmosphere Near real-time Capability for EOS (LANCE) system (*https://earthdata.nasa.gov/lance*), part of NASA's Earth Observing System Data and Information System (EOSDIS).

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

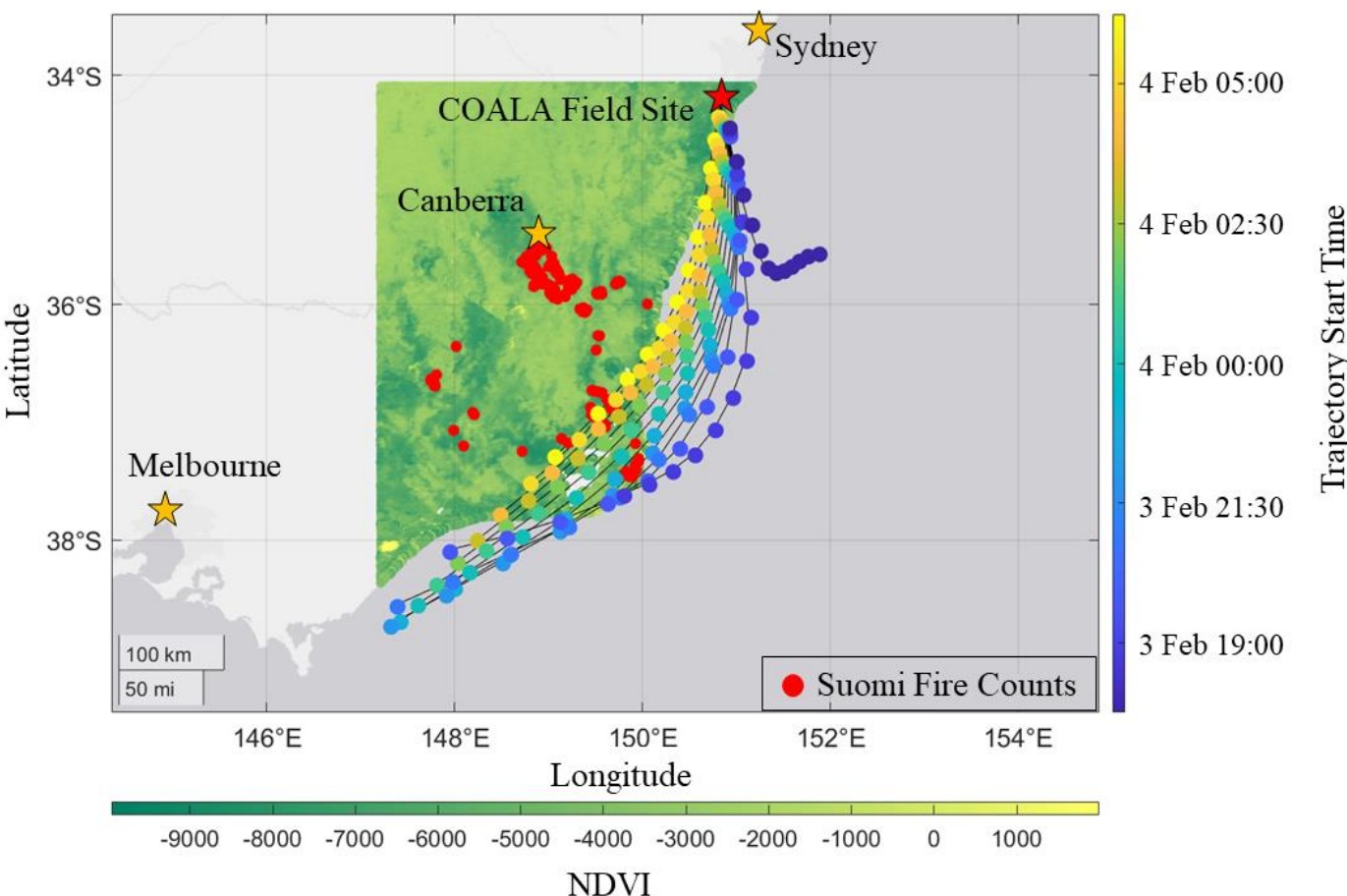

**Figure 1: Active fires from 1-5 Feb. 2020 and their proximity to the COALA-2020 field site. NDVI is plotted at 250m resolution from the MOD13A1 dataset acquired by measurements via the MODIS Terra satellite. Pixels have been filtered to contain cloud coverage less than 30% and VI Usefulness bits indicating top two tiers of data quality. Fire counts are plotted using the VNP14IMGTDL_NRT data from Suomi VIIRS satellite imaging overlaid with HYSPLIT back trajectories. Each tail represents a trajectory 12 h prior to reaching the site and is colored by its starting time. Circles indicate 1 h intervals moving backwards from the start time.**

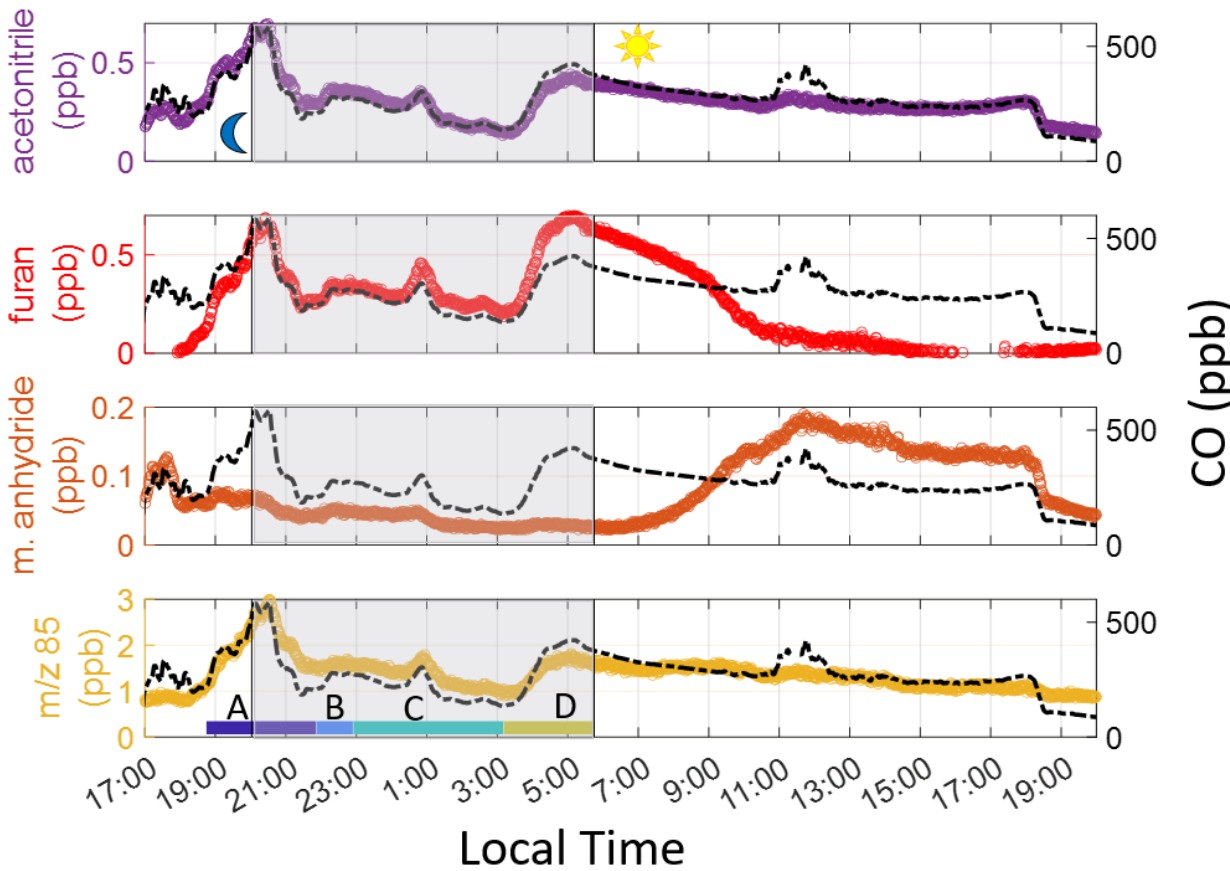

**Figure 2: VOCs and CO on 3 – 4 Feb 2020 with the shaded area representing sunset to sunrise. The peak in CO after sunset (start of gray-shaded area) is used to denote the beginning of the smoke event. We limit our analysis to sunrise on the following day. The color labels A-D indicate individual times used to calculate ERs (see section 5.2 in main text). m/z 85 in the bottom time series indicates the sum of furanone and cis-2-butenediol.**

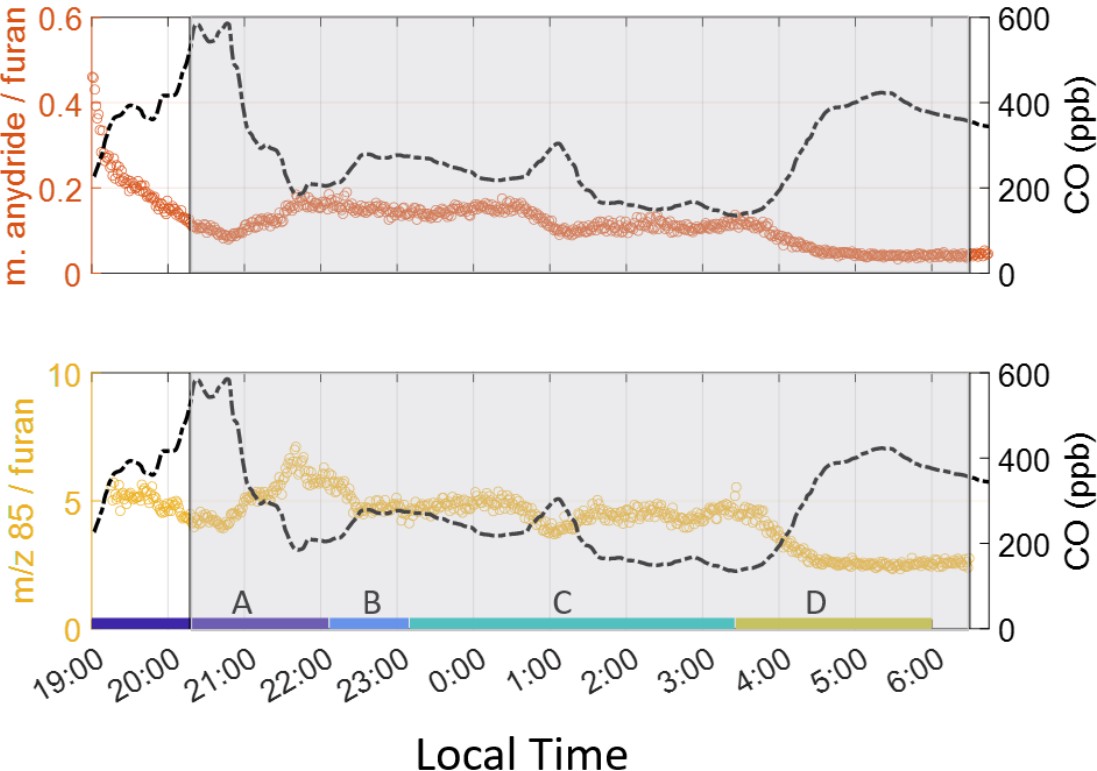

**Figure 3: Product-to-reactant ratio for furan oxidation products. Both ratios indicate the period just before sunrise is least oxidized. Again, the color labels A-D indicate individual times used to calculate ERs. m/z 85 indicates the sum of furanone and cis-2-butenediol.**

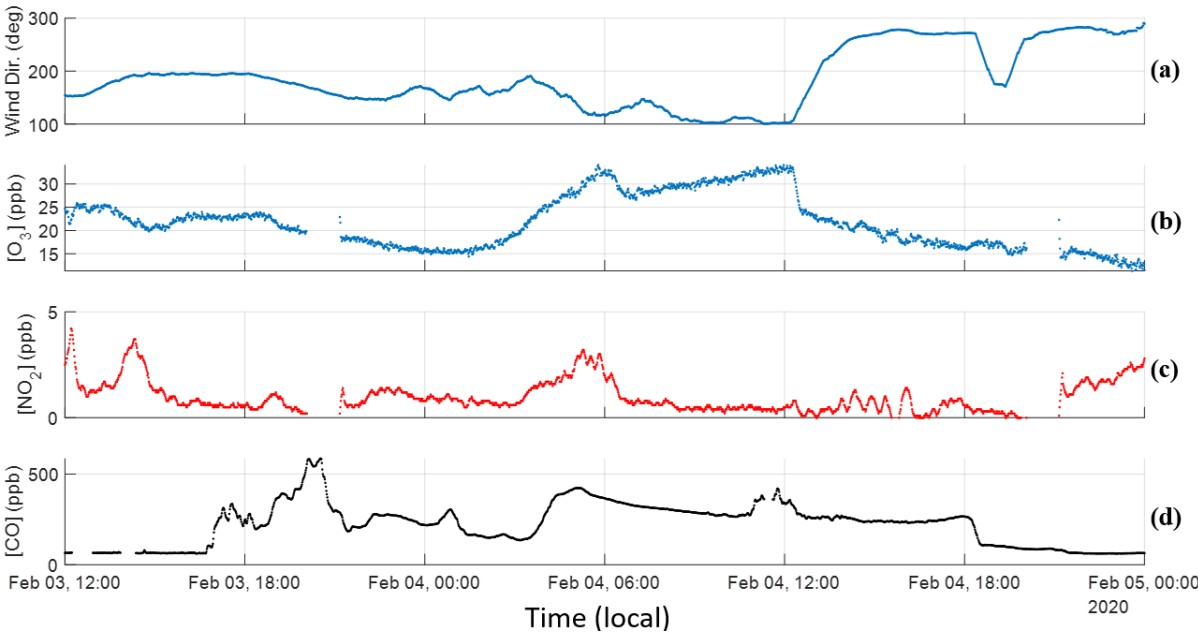

Figure 4: Time series for O₃, NO₂, CO, and wind direction. (a) Wind direction is read as true north is 0∘ and east as 90∘. (b) O₃ trends well with CO until sunrise occurs, wherein BBVOC+OH oxidation combined with biogenic VOC emissions led to daily production. The close trend with CO over the nighttime indicates transport rather than local formation. (c) NO₂ also shows a similar trend but upon sunrise begins to negatively correlate with CO and O₃. (d) CO smoke tracer provide as time series reference.

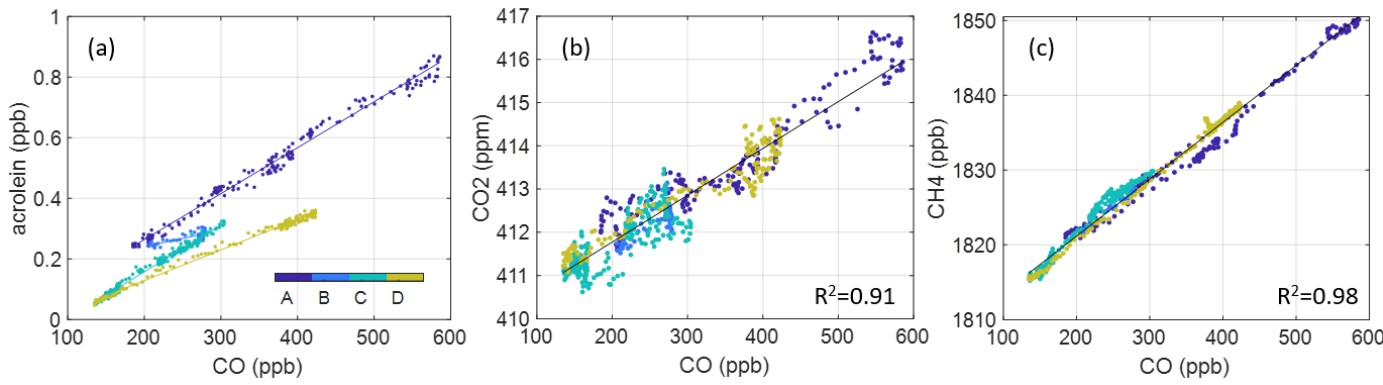

Figure 5: Example ER analysis (a) using acrolein, wherein the smoke event is partitioned into 4 periods over the evening. Average ERs (slopes) from periods A-C agree closely with those in the freshest portion of the plume (D). Panels (b) and (c) show the singular ERs derived for CO₂ and CH₄ using the entire nighttime dataset (A-D).

**Table 1: EFs (g kg⁻¹) derived in this work compared to 2 studies conducted in the same or near temperate Australian forests, 2 US-based aircraft campaigns sampling western temperate US fuels, and 1 study reporting EFs across geographically distant temperate forests.[*] Again, m/z 85 indicates the sum of furanone and cis-2-butenediol.**

| | | | Biome Location | | | | | |
|---|---|---|---|---|---|---|---|---|
| | | | AU | AU | AU | US | US | Temperate Forests |
| Compound | Formula | m/z | This Work | Guerette et al. (2018) | Lawson et al. (2015) | Liu et al. (2017) | Permar et al. (2021) | Akagi et al. (2011) |
| Methanol | $CH_4O$ | 33.00 | $2.01 \pm 0.58$ | $3.0 \pm 0.5$ | $2.07 \pm —$ | $2.45 \pm 1.43$ | $1.50 \pm 0.39$ | $1.93 \pm 1.38$ |
| Acetonitrile | $C_2H_3N$ | 42.03 | $0.16 \pm 0.03$ | $0.70 \pm 0.10$ | $0.25 \pm —$ | $0.25 \pm 0.13$ | $0.31 \pm 0.15$ | — |
| Acetaldehyde | $C_2H_4O$ | 45.03 | $0.57 \pm 0.20$ | $1.20 \pm 0.30$ | $0.92 \pm —$ | $1.64 \pm 0.52$ | $1.70 \pm 0.43$ | — |
| Acrolein | $C_3H_4O$ | 57.03 | $0.23 \pm 0.08$ | — | — | — | $0.40 \pm 0.18$ | — |
| Acetone | $C_3H_6O$ | 59.05 | $0.55 \pm 0.28$ | $0.80 \pm 0.20$ | $0.54 \pm —$ | $1.13 \pm 0.82$ | $0.84 \pm 0.22$ | — |
| MVK+MACR | $C_4H_6O$ | 71.05 | $0.18 \pm 0.03$ | $1.0 \pm 0.30$ | $0.38 \pm —$ | $0.33 \pm 0.06$ | $0.39 \pm 0.15$ | — |
| Benzene | $C_6H_6$ | 79.05 | $0.25 \pm 0.08$ | $0.39 \pm 0.07$ | $0.69 \pm —$ | $0.43 \pm 0.12$ | $0.50 \pm 0.14$ | — |
| m/z 85 | $C_4H_4O_2$ | 85.03 | $0.83 \pm 0.27$ | — | $0.57 \pm —$ | $0.39 \pm —$ | $0.32 \pm 0.11$ | — |
| Methyl-propanoate | $C_4H_8O_2$ | 89.06 | $0.07 \pm 0.03$ | — | — | — | $0.081 \pm 0.036$ | — |
| Maleic-anhydride | $C_4H_2O_3$ | 99.00 | $0.05 \pm —$ | — | — | — | $0.14 \pm 0.072$ | — |
| Methyl-methacrylate | $C_5H_8O_2$ | 101.06 | $0.07 \pm 0.04$ | — | — | — | $0.11 \pm 0.045$ | — |
| Benzaldehyde | $C_7H_6O$ | 107.05 | $0.05 \pm —$ | — | — | — | $0.04 \pm 0.026$ | — |
| C8-aromatics | $C_8H_{10}$ | 107.09 | $0.08 \pm 0.05$ | $0.11 \pm 0.03$ | $0.26 \pm —$ | $0.15 \pm 0.004$ | $0.21 \pm 0.08$ | — |
| C3-benzene | $C_9H_{12}$ | 121.10 | $0.07 \pm 0.06$ | — | $0.27 \pm —$ | — | $0.069 \pm 0.031$ | — |
| Creosol | $C_8H_{10}O_2$ | 139.08 | $0.05 \pm —$ | — | — | — | $0.14 \pm 0.11$ | — |

[*] Dashes indicate either EF or EF variability not reported in study.

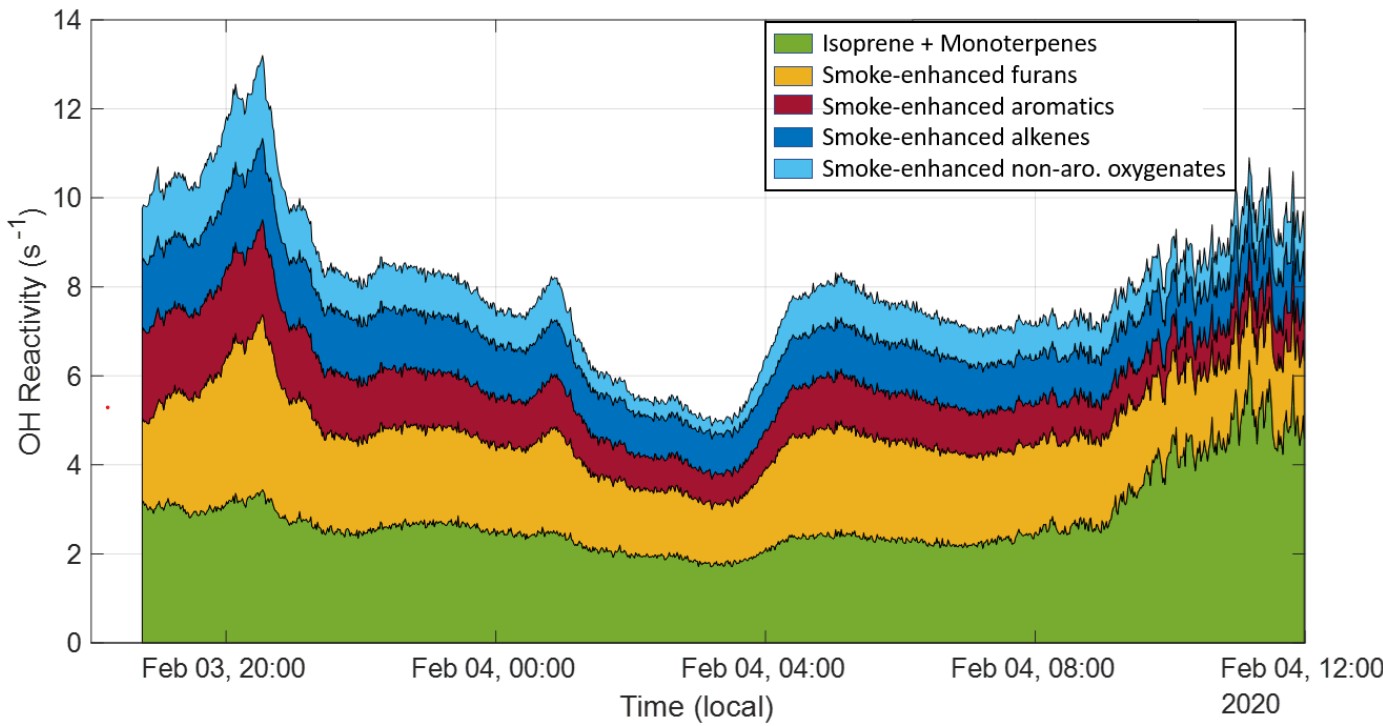

**Figure 6 – Selected compounds with significantly high smoke-related enhancements are grouped into categories of varying reactivity based on known reactivity groups, except for the "isoprene + monoterpenes" group which is the sum of isoprene (m/z 69) and monoterpene (m/z 137) reactivities. This captures every compound included in this OH reactivity calculation.**