# Peer review of "Measurement Report: Observations of long-lived volatile organic compounds from the 2019-2020 Australian wildfires during the COALA campaign"

_Atmospheric Chemistry and Physics, 2021_

## Author Comment (AC1)

The authors thank the reviewers for their helpful comments and suggestions. The original comments are below in black, and our responses follow in green.

Notably, in response to the reviewers' concerns about the scope of the analysis, we have incorporated a broader range of gas-phase measurements and a more in-depth discussion. We resubmitted the manuscript as a Measurement Report. As our measurements are some of the only in-situ observations of atmospheric composition during the historical 2019-2020 Australian Black Summer, we believe this represents an important contribution to the literature that is within the scope of ACP.

Though not discussed in responses below, our revised manuscript has removed some compounds and their EFs due to insufficient PTR mass resolution discovered in our more in-depth data analysis.

**Referee #1:**

**General Comments:**

The authors present a set of 21 emission factors (EFs) for longer-lived VOCs measured using a Proton-Transfer-Reaction Time-Of-Flight Mass Spectrometer (PTR-TOF-MS) during an opportunistically sampled wildfire plume strike.

This may be the first deployment of PTR-TOF-MS in Australian temperate forest wildfire smoke, however the need to restrict the analysis to the least reactive species due to the age of the plume (~8h) reduces the number of reported species to 21 (down from 150 identified species) and therefore the added value of using this instrumentation is lost, with relatively few (9) new species quantified for the ecosystem.

We agree that the number of new species quantified for this ecosystem is regrettably small. However, we also believe that there is value in repeated observations of previously reported species, for investigation of fire-to-fire variability.

The authors use the ratio of maleic anhydride to furan to assess OH oxidation in the plume and the ratio of (cis-2-butenediol + furanone)/furan to assess $NO_3$ oxidation. The use of (cis-2-butenediol + furanone)/furan as an indicator is new but is confounded by the fact that the oxidation products are emitted by the fire itself.

Thank you for this remark. As m/z 85 is emitted in the combustion process, we restrict ourselves from calculating an exact age and instead use this marker for relative comparisons over different periods of the night in determining which among them is least oxidized. Given the well correlated time series trends of m/z 85 and maleic anhydride ratios, and the corresponding decrease in both for the nighttime period D, we are confident that these compounds are elucidating a comparative decrease in oxidation of the plume.

The manuscript then includes a comparison with EFs reported by a selected number of other studies.

As it is, the manuscript is a reasonably well-presented data paper, but the lack of a discussion means that if falls outside of the scope of ACP. As the paper is short, I recommend adding a discussion of the findings in a revised version.

> We have expanded the discussion part of the manuscript and included a wider range of observations in the new manuscript. We believe the work presented falls within the "Measurement Report" category of ACP.

**Specific comments/questions:**

**Introduction:**

Lines 27-28: There are other references to include to support this statement, including some studies that have a focus on Australia.

> Additional citations have been added to this section, several of which are focused on Australia. This change is reflected starting at line 34 in the revised document.

Lines 35-36: Could you expand on this? And add references? There are so many papers describing this event. Anything about the blanketing smoke would seem relevant in the context of this manuscript. In reference to the below passage:

> We have added additional discussion and references describing the Black summer event to provide a fuller picture of this global issue. These changes are reflected starting on line 46.

Line 36-37: Please add references. In reference to the passage below:

> The manuscript and citations have been altered to incorporate this comment.

Line 37-38: Akagi et al 2013 and Burling et al 2011 are examples of measurements of EFs in North American temperate forests, not of CTMs using them. Do you have examples of CTMs using North American values in an Australian context and getting poor results as asserted on line 37? In reference to the line below:

> Models run in Lawson et al. (2017) have incorporated EFs derived from those same studies by Akagi et al. 2013 or Burling et al. 2011. The manuscript has been updated to expand more on this statement. Changes in the manuscript incorporating this begin on line 55.

**Section 2:**

Lines 87-88: Was the instrument calibrated after the event as well? Was there much drift in between?

> The PTR-ToF-MS was not calibrated directly after the event, but calibrations were conducted 2-3 times per month, with the nearest one being a week later. Drift in instrument sensitivity over the course of

the campaign was within 15% of the first recorded sensitivities. Line 129 has been included in the manuscript to note this.

Line 94: Please add a table (in the supplementary info) of all the reported species, what m/z they were measured at, whether they were contained in the calibration cylinders and an indication of uncertainty for each:

Table S2 is now included in the supplement to incorporate this comment.

A campaign such as COALA would have had access to O3 and NOx measurements. Could you add these species to the figures in section 3?

A discussion of $NO_2$ and $O_3$ have been included in the main body of the updated manuscript.

**Section 3**

Figures 2 and 3: Please explain that m/z 85 is the sum of cis-2-butenediol and furanone in the caption.

Thank you, this was an oversight. Captions for these figures have been updated.

Line 146: winds from the north? aren't the fires to the south of the site? Can you clarify?

The reviewer is correct. The fires are in the southeast corner of the continent. The winds are blowing to the north, not from it. The manuscript has been updated for clarification at line 224.

**Section 4**

This section needs clarification.

Line 158: "when passing over the active fires ~25 km to the south, near Canberra" 25 km to the south of what? There are no trajectories going over Canberra in Figure 1.

Line 160: "the intensity of the fires near Canberra" Again, none of the trajectories go over the fires near Canberra. I can see how different fires have been sampled (blueish colour right on the coast, yellowish colour a little inland).

The mention of the Canberra fires is made to discuss potential sources of influence on the plume we sampled. We have rectified line 224 and included forward HYSPLIT trajectories starting at the point of the Canberra fires to clarify this section.

Line 163: 25km seems too small a distance

The "25 km" was in reference to the distance between the fire cluster closest to the center of the city. The updated manuscript has clarified this discussion.

**Section 5**

Line 182: The use of standard linear regression is not appropriate here. Use something appropriate like reduced major axis regression.:

Thank you for this comment. All emission ratios and emission factors have been recalculated using the reduced major axis regression. The main body of the manuscript has been accordingly updated to convey this information. As mentioned previously, a high correlation coefficient ensures that values across regressions will be largely consistent, and resultingly our conclusions have not changed.

Line 199: "allow us to report ERs for…" Are these ERs tabulated anywhere? It would be nice to see them, maybe in the supplementary info.:

Thank you for the suggestion. Table S3 is now included in the supplement and details the ERs for each compound.

Table 1: Add the m/z at which the species were measured. Also, isn't Furanone the sum of furanone and cis-2-butenediol?

Table 1 has been updated to include a new column for masses and the name rectified. Denoting the m/z 85 EF as furanone was an oversight and has been amended. Thank you for pointing this out.

General comments on Section 5

Considering that Permar et al report potential uncertainties of up to a factor of 2 (see Figures 2 and 5 of their paper) and that similar uncertainties apply to this study (one plume sampled) and most of the other studies that are included in the comparison, it seems likely that any 'discrepancies' of up to a factor of 2, 3 or even 4 are actually not significant. The only species that then warrant commenting upon are MVK+MACR, acetonitrile, propene, potentially methyl methacrylate?

The text has been updated in the abstract, discussion, and conclusion to reflect this comment and we thank the referee for mentioning this. Many of our compound EFs fall within a factor 2.5, and a substantial majority are within a factor of 5. The text has been updated to indicate this range as reasonable to good agreement, rather than just 'mixed.' While it may seem that this means our results are not "novel", we highlight that additional observational confirmation of previously quantified EFs is valuable.

Can you calculate the 'modified combustion efficiency' of the plume sampled?

We thank the reviewer for this suggestion. We calculate an average MCE using the $CO_2$ and CO ERs determined for this manuscript, again because this accounts for background values:

$ER_{CO2/CO}$ = 10.83, so MCE = 10.83 / (10.83 + 1) = 0.92

This MCE indicates a less efficient combustion that is an even mixture of smoldering and flaming. The manuscript has been updated to include information regarding MCE.

The manuscript needs a discussion section. EFs were measured, they are in rough agreement with other EFs. So what? What makes this more than a data paper? Much of the east coast of Australia was blanketed in smoke for weeks, do your findings shed any light on anything related to this? The species you report on are 'longer-lived' – what are their potential impact on downwind chemistry? Did you see anything interesting in ozone or particulates when the sun rose on the second day? Could you use your oxidation indicators to 'wind back the clock' and determine EFs for more species? As it stands, this manuscript would be better off published elsewhere.

In this revised submission, we have attempted to more fully quantify the gas phase aspects of this smoke event through discussion of $O_3$ and $NO_2$ time series, as well as identification and calculations of the dominant contributing compounds to OH reactivity.

Important findings are that we see a $NO_x$-limited plume that is transporting $O_3$ to the field site rather than promoting local production. Additionally, when compared to a transported, nighttime plume from Liang et al. (2022) which occurred in an urban setting, we see opposite time series trends between $NO_2$ and maleic anhydride in a daytime oxidized plume. We also quantify the major compounds contributing to OH reactivity over the course of the evening, finding comparable reactivity values (range of 3.15 – 9.83 $s^{-1}$ with an average of 5.25 $s^{-1}$) to Liang et al. (2022), as well as similar classes of compounds (furans, alkenes, and phenols) making up the majority of remaining OH reactivity.

**Technical Corrections:**

- Line 42: delete 'in': "higher than those in measured …"
- Line 61: replace 'select' with 'selected'
- Line 66: insert "the" in "30 km to the northwest"
- Line 83: "assist pump"? Wouldn't it be clearer to say something like "air was pulled through the inlet at a flow rate of 3 SLPM for a residence time of 2.5s and the PTR-TOF-MS sampled at X flow rate from this bypass flow"?
- This line has been updated for clarity.
- Line 184: Replace 'the' with 'that'?: "We find the using only the freshest…"

All technical corrections below have been implemented in the updated main text:

**Referee #2:**

**Major comments:**

General Comments:

Comment #1: Basis for analysis - I have several basic questions around the rationale and choice of the primary methods for the analysis. For example, I am not sure I fully followed why the sunset to sunrise time was selected for analysis. While the concentrations measured right before sunrise on Feb 4 are likely to be the least affected by O3 and OH driven oxidation chemistry (assuming most of the emissions are picked up during transport over the previous night), it is unlikely that the samples measured at sunset on the previous day on Feb 3 can be assumed to be unoxidized.

> We do not assume that the samples measured on sunset of 3 Feb are unoxidized. We instead investigate the entire period when we were sampling a smoke impacted plume (as denoted by enhanced furan concentrations) and compare the relative level of oxidative processing throughout the smoke-enhanced period. Using HYSPLIT trajectories, we determine that the plume had traveled for ~8 hrs, and the "freshest" portion of the plume (period D, lowest OVOC/VOC ratios) was transported overnight, when $O_3$ and OH oxidation chemistry was minimal.

The opposite would be true for oxidation via $NO_3$ although species that are oxidized by all 3 radicals (i.e., $O_3$, OH, and $NO_3$) would suffer from several layers of confounding, since the oxidant exposures are not known.

> We agree that oxidation could be occurring from both $NO_3$ and OH for periods A-C and examine this in Section 6.2 (previously 5.2) as well as the supplement (Fig. S8). Comparisons of the solely $NO_3$ oxidized period D to the average of periods A-D show that there is little variability across them. We assume $O_3$ oxidation is slower compared to $NO_3$ oxidation.

A case in point is that the change in the maleic anhydride to furan ratio (Figure 3 top panel) to a lower value coincides with the shift in where the air parcels are coming from, questioning the appropriateness to use this ratio to determine the level of aging in that air parcel. In fact, the ratio decreases even further around 3 am on Feb 4 indicating that the air parcel at or after 3 am might represent the freshest plume possible.

> We agree, and this conclusion is fundamental to our analysis. Air parcels consistently come from the southeast, as conveyed in Fig. 1 via HYSPLIT trajectories. The age marker ratios indicate that the freshest period is (D) (Fig. 3).

That the authors do not directly account for oxidation of the direct emissions of the species measured and production from the oxidation of their precursors, further adds to my concern.

> We apologize if this was not made clear in the manuscript, but we account for the oxidation of directly emitted species using HYSPLIT outputs and limiting compounds to those with sufficiently low $NO_3$ reaction rates (as well as the additional comparisons mentioned earlier in this comment). We are encouraged given the good agreement of our compounds with the comparative studies. We do acknowledge that there are sources for secondary formation of methanol and acetaldehyde (Holzinger et al., 2005), and state that our m/z 85 EF is likely inflated due to transport time and plume oxidation. Acrolein is accounted for in further comments below.

For instance, aren't furans oxidation products of aromatic oxidation chemistry too? How is this accounted for?

> This is an excellent point. We do not account for aromatic contributions to furan oxidation products because our use of maleic anhydride and m/z 85 are not contingent on precise values and are not used to calculate an absolute age. If both aromatic and furan oxidation are consistently contributing to m/z 85 or maleic anhydride production (for which we know furan is from our time series), then this doesn't affect the conclusions we draw using the product-to-reactant ratios as period D will be, relatively, less oxidized than the preceding periods.

Another objection with this analysis is that it seems impractical to analyze these specific species concentrations to construct EFs and emissions ratios (ERs), given that the air parcel may have picked up emissions over 100s of kilometers from fires at very different stages of burning and varying environmental conditions and the air parcel studied would have very well mixed with other air parcels arising from different source regions.

> Thank you again for these observations. Further analysis was conducted to address these points. Analysis is included in the supplement in Section S3. Text in the manuscript has also been updated to address this in Section 4, line 224.
>
> In short, we conclude through HYSPLIT forward trajectories that the plume sampled was almost entirely from the southeast fires shown in Fig. 1, with little mixing from the large fires active to the south of Canberra. This leaves only the potential for a few small, sparse fires to contribute. Furthermore, we observe monthly average rainfall in the region of these southeast fires, as well as wind speeds before, during, and after the smoke event and conclude that these burns were happening under similar conditions.
>
> Additionally, setting the $R^2>0.5$ for our ERs removes the need for background corrections that could have been influenced from neighboring sources. High correlations in our ERs maintain that influences from other sources (i.e. traffic for benzene) are accounted for.

The primary problem is that the measurement site is too far from the source of the fires. I am not convinced that this dataset can be used to infer EFs and ERs in a robust manner and the mixed agreement with previous measurements from the same region is hence not surprising.

> What this work considers to be "mixed" or "good" agreement has been adjusted in line with comments from the other referee. Overall, the results here show reasonable agreement with Lawson et al. (2015), Liu et al. (2017), and Permar et al. (2021). Updates have been made to the text in the abstract, discussion, and conclusions. We find these results to be encouraging in regards to their validity. We have since added more discussion regarding a time series analysis of $NO_2$ and $O_3$, as well as investigated the transported OH reactivity measured at the field site. We find reactivity values matching those from other literature (in the manuscript) and see similar classes of compounds contributing most of this reactivity. We believe the manuscript now fulfills the specifications for a measurement report.

Minor comments:

1. Abstract: It would be good to describe the comparisons with earlier work quantitatively. That way the reader can understand the difference between 'mixed' and 'good'.

Thank you for making this point. Our conclusions have been updated to be more quantitative in the abstract, discussion, and conclusion sections of the text.

2. Page 1, line 28: Is there a reference newer than Liu et al. (2010)?

   Additional references have been added to incorporate this comment.

3. Page 2, lines 41-43: Is there a mechanistic reason for higher EFs for these species in Australia?

   The text has been updated to incorporate this comment starting on line 62. In the Guérette et al. (2018) paper, they postulate that fuel type is the dominant reason for higher EFs, especially for acetonitrile (much of the total understory mass of the temperate Australian forests contain fuel types with higher nitrogen content like acacias). This is also mentioned in the "Calculating Emission Factors" section.

4. Page 3, line 82: What is the latest understanding on sampling artifacts for VOCs using PTFE tubing as a function of volatility, functional group, etc.?

   Generally, for reductions in wall losses, one maintains as short a line as possible with as thin an inner diameter (ID) as possible. Typically, this means line lengths are between 5 – 15 m with IDs usually at 1/8" or less. We employed ¼" outer diameter PTFE tubing with an 1/8" ID, which connected to our PTR-ToF-MS through a 1/16" PEEK tubing section (both are non-reactive plastics that aid in wall loss prevention). Furthermore, high flow rates are preferable in the range from 2 – 20 SLPM, as are insulated or even heated inlets (usually heated to ~50 C). Our instrument heats its PEEK tubing section to 60 C.

   Heavier compounds with high carbon counts like monoterpenes experience wall loss with literature from Pagonis et al. (2017) noting the most intense partitioning from $C_8 – C_{14}$ 2-ketones and $C_{11} – C_{16}$ 1-alkene groups.

5. Figure 1: Show Sydney and Melbourne too to orient the reader who may not be familiar with that part of the world?

   Fig. 1 has been updated to incorporate this comment.

6. Page 5, lines 110-117: I am not convinced that one can use the maleic anhydride to furan ratio to quantify the absence of oxidation for two reasons. Gkatzelis et al. (2020) measured these ratios:

   (i) much closer to the fire than this work. Can you arrive at the same conclusion if you use other ways to assess oxidation, e.g., MEK+MVK:isoprene, toluene:benzene? Note that these may not be the right VOC ratios to be used in this work.

   Thank you for this observation. From our colleague's paper (Simmons et al. (2022) (submitted)) in Fig. 4(a), a ToF-ACSM was used to measure what fraction of $PM_1$ has a mass-to-charge ratio of 44 ($f_{44}$). ACSM results show that our smoke event achieves a lower fractional content at $f_{44}$ as the night progresses and indicates that the plume is at its youngest around the timeframe that corresponds to our period (D), thus corroborating our analysis. Additionally, we are only quantifying the absence of oxidation in regards to other time periods of the smoke event. We acknowledge this plume has

experienced significant oxidation and use the transport time from the HYSPLIT back trajectories to gauge this.

Regarding other tracers, toluene/benzene won't be as informative as the compounds observed in this study as both have low ERs with wildfire smoke and possible biogenic influences. Furthermore, because this smoke event happened over the evening, using an isoprene-based age marker is not as informative as isoprene is largely depleted in the nighttime, has no correlation to wildfire smoke per our results, and this biogenic oxidation marker assesses oxidation from the OH radical only.

(ii) were representative of emissions from western US fuels.

The time series shows low nighttime concentrations during the smoke event, indicating little to no emission, and then an increase coinciding with sunrise and subsequent furan + OH oxidation. The time series trend gives us significant confidence that this is maleic anhydride being measured. Furan undergoing reactions with OH will lead to maleic anhydride production.

7.  Page 5, Line 133: To me both acetonitrile and furan seem to go up around the same time at approximately 18:00 hours on Feb 3. So, I don't agree with this opening sentence. Why can't the base acetonitrile level between 17:00 and 18:00 on Feb 3 be background?

    Thank you for this comment. The text has been updated to address this point. Additionally, acetonitrile is a strong smoke tracer because of its virtually zero background concentration (especially in a forested setting), so we would not anticipate measuring it otherwise.

8.  Page 5, lines 118-143: I think I understand (but disagree; see above) the OH argument made here. But I am not sure I followed the $NO_3$ argument. To me, the m/z 85 to furan ratio is relatively constant over the entire period shown. As pointed out earlier for the maleic anhydride to furan ratio, the dip in the m/z 85 to furan ratio comes about at 3 am on Feb 4.

    We apologize if this point is unclear. The concentration dip in both m/z 85 and maleic anhydride relative to furan at 03:00 is a central point to our conclusions. We believe this effectively shows that after 03:00 we see a relatively less oxidized smoke plume, which is lower by a factor of 1.6 – 2.8 for the m/z 85 / furan time series. Additionally, it should be noted that all compounds selected have lifetimes that are suited to the transport timeframe.

9.  Figure 4(a): The slopes are different and decreasing with time suggesting acrolein is produced in the samples measured earlier? This doesn't seem to have been discussed in Section 5.2 clearly.

    We apologize if the figure is unclear: the slopes do not consistently decrease with time. While slope B > slope A, C is then the largest slope, and D is nearly the same as A. Additionally, acrolein has a known production source from alkene + OH reactions, but this is deemed negligible relative to wildfire emissions (O'Dell et al., 2020). It also has known production source in the nighttime from 1,3-butadiene + $NO_3$ reactions, but less than 5% of these reactions yield acrolein, so we deem this negligible as well (Skov et al., 1992).

10. Figure 4: Are there examples of species where the correlations with CO are much more scattered or segregated with time than the species shown in the main text? This could be instructive on where this method fails. This information would also be useful to include in the SI for all species.

Thank you for this point. Species that this method would fail to capture when considering a prolonged transport time would be those that have significant dependencies on MCE, are highly reactive, and do not have comparatively large EFs. This means what little concentrations were emitted would react away quickly and their emissions would be contingent on low-efficiency combustion. An example would be pyrrole, which often correlates well with CO during wildfires. However, our measurements do not exhibit this trend and show only very low levels of pyrrole, most likely because it has been reacted away. We've included a statement regarding this in the main text on line 301.

11. Sections 5.2-5.3: To me, there needs to be a discussion of what the background concentrations of these species are and its variability when the air parcel being sampled is not influenced by biomass burning. Also, why are the data not corrected for background concentrations?

    We apologize if we did not make this point appropriately clear. This method of ER derivation is chosen because it precludes the need for calculating excess mixing ratios (it implicitly accounts for background concentrations). The main text has been amended to emphasize this point in Section 6.2. Additionally, compounds used in calculating OH reactivity are background corrected (discussed further in Section 6.4). The correlations for a significant majority of our ERs are also well above 0.5 indicating strong association with wildfire emissions. Further details regarding influence on plumes from surrounding sources are discussed in Simmons et al. (2022) (submitted). The text has been updated to include this reference.

12. Table 1: Can these data be compared to laboratory studies and what could be learned from that comparison? Recent examples include Stockwell et al. (ACP, 2015), Hatch et al. (ACP, 2018), Koss et al. (ACP, 2018), and Sekimoto et al. (ACP, 2018).

    Thank you for this comment. We compared EFs with both Stockwell et al. (2015) and Koss et al. (2018) in the initial iteration of this manuscript. We have added to the supplementary a section looking at a comparison of our EFs with those from Koss et al. (2018) (specifically those averaged over all fuel types) and the EFs from Stockwell et al. (2015). For the latter EFs, we calculated an average from the fuel types in the study that would be present in a purely temperate setting (i.e. no savannah fuel types or emissions from cooking). This information is now displayed in Fig. S10.

    Essentially, there is excellent agreement with our EFs (all compounds within uncertainty) and those presented in the other two studies. This indicates the ability to employ averaged, lab-based values across geographically separate, but analogous biomes.

Can MCE explain differences in the EFs between this and earlier work?

    We thank the reviewer for this suggestion. We've calculated an average MCE using the $CO_2$ and CO ERs determined for this manuscript, again because this accounts for background values:

    $ER_{CO2/CO}$ = 10.83 ppbCO$_2$ ppbCO$^{-1}$, so MCE = 10.83 / (10.83 + 1) = 0.92

    This MCE indicates a less efficient, even mixture of smoldering and flaming combustion. The general trend of smaller EFs in this work as opposed to those reported in other studies implies that MCE is not a singly explanatory variable for any discrepancies, as this MCE is not particularly low or high compared to values found in Lawson et al. (2015), Koss et al. (2018), or Permar et al. (2021). The manuscript has been updated to include information regarding MCE.

It would be good to also show these comparisons on a scatter or bar plot.

A scatter plot of all EFs and respective uncertainties has been provided in the supplement (Fig S9) for better visual conveying of information.

Finally, could the dominant fuel be highlighted in this table across the different studies?

We cannot determine the dominant fuel for all reports as Permar et al. (2021) doesn't report a dominant type. For Australian field campaigns, the dominant fuel types would be various species of Eucalyptus (noted in Section 2.1 of this work).

13. Do the ERs between studies (compared in Table 1 and Section 5.3) align better if one used a different tracer instead of CO (e.g., acetonitrile)?

The studies we compare our results with dominantly use CO for an emission ratio and ultimately this remains the most apt tracer to use for ERs. Guérette et al. (2018) uses $CO_2$ for several compound ERs, but we can only compare with one of them (benzene). Following are the results: $ER_{benzene/CO2} = 8.26 \times 10^{-5}$ +/- $9.13 \times 10^{-6}$ with an $R^2=0.83$ (3 of 4 periods satisfy $R^2$ criteria). This is compared to the Guérette et al. (2018) value at $ER_{benzene/CO2} = 1.4 \times 10^{-4}$ +/- $2 \times 10^{-5}$. This has a relative difference of -41.02%. This indicates poor agreement.

14. **Section 6: Do the effective EFs for the other more reactive species measured by the PTR-ToF-MS conclusively show the role of oxidation during day and night?**

Provided are 4 compounds highly reactive to both OH and $NO_3$, and that are well known reactive tracers for wildfire combustion (cresol, guaiacol, methylfuran, 2,5-dimethylfuran) to indicate the role of oxidation. They exhibit values lower by an order of magnitude compared to Permar et al. (2021). These compounds lack counterpart values from Australian biomes. Effectively, this is showing that highly OH reactive species are reacted away prior to reaching the field site, and as a result exhibit substantially lower EFs that campaigns sampling fresh plumes.

| Compound | EF (g kg$^{-1}$), this work | EF (g kg$^{-1}$), Permar et al. (2021) |
|---|---|---|
| Cresol | 0.033 | 0.23 |
| Guaiacol | 0.042 | 0.27 |
| Methylfuran | 0.0595 | 0.28 |
| 2,5-dimethylfuran | 0.0146 | 0.20 |

**References:**

Guérette, E. A., Paton-Walsh, C., Desservettaz, M., Smith, T. E. L., Volkova, L., Weston, C. J., and Meyer, C. P.: Emissions of trace gases from Australian temperate forest fires: emission factors and dependence on modified combustion efficiency, Atmos. Chem. Phys., 18, 3717-3735, 10.5194/acp-18-3717-2018, 2018.

Holzinger, R., Williams, J., Salisbury, G., Klüpfel, T., de Reus, M., Traub, M., Crutzen, P. J., and Lelieveld, J.: Oxygenated compounds in aged biomass burning plumes over the Eastern Mediterranean: evidence for strong secondary production of methanol and acetone, Atmos. Chem. Phys., 5, 39-46, 10.5194/acp-5-39-2005, 2005.

Koss, A. R., Sekimoto, K., Gilman, J. B., Selimovic, V., Coggon, M. M., Zarzana, K. J., Yuan, B., Lerner, B. M., Brown, S. S., Jimenez, J. L., Krechmer, J., Roberts, J. M., Warneke, C., Yokelson, R. J., and de Gouw, J.: Non-methane organic gas emissions from biomass burning: identification, quantification, and emission factors from PTR-ToF during the FIREX 2016 laboratory experiment, Atmos. Chem. Phys., 18, 3299-3319, 10.5194/acp-18-3299-2018, 2018.

Lawson, S. J., Keywood, M. D., Galbally, I. E., Gras, J. L., Cainey, J. M., Cope, M. E., Krummel, P. B., Fraser, P. J., Steele, L. P., Bentley, S. T., Meyer, C. P., Ristovski, Z., and Goldstein, A. H.: Biomass burning emissions of trace gases and particles in marine air at Cape Grim, Tasmania, Atmos. Chem. Phys., 15, 13393-13411, 10.5194/acp-15-13393-2015, 2015.

Liang, Y., Weber, R. J., Misztal, P. K., Jen, C. N., and Goldstein, A. H.: Aging of Volatile Organic Compounds in October 2017 Northern California Wildfire Plumes, Environmental science & technology, 2022.

Liu, X., Huey, L. G., Yokelson, R. J., Selimovic, V., Simpson, I. J., Müller, M., Jimenez, J. L., Campuzano-Jost, P., Beyersdorf, A. J., Blake, D. R., Butterfield, Z., Choi, Y., Crounse, J. D., Day, D. A., Diskin, G. S., Dubey, M. K., Fortner, E., Hanisco, T. F., Hu, W., King, L. E., Kleinman, L., Meinardi, S., Mikoviny, T., Onasch, T. B., Palm, B. B., Peischl, J., Pollack, I. B., Ryerson, T. B., Sachse, G. W., Sedlacek, A. J., Shilling, J. E., Springston, S., St. Clair, J. M., Tanner, D. J., Teng, A. P., Wennberg, P. O., Wisthaler, A., and Wolfe, G. M.: Airborne measurements of western U.S. wildfire emissions: Comparison with prescribed burning and air quality implications, Journal of Geophysical Research: Atmospheres, 122, 6108-6129, https://doi.org/10.1002/2016JD026315, 2017.

O'Dell, K., Hornbrook, R. S., Permar, W., Levin, E. J. T., Garofalo, L. A., Apel, E. C., Blake, N. J., Jarnot, A., Pothier, M. A., Farmer, D. K., Hu, L., Campos, T., Ford, B., Pierce, J. R., and Fischer, E. V.: Hazardous Air Pollutants in Fresh and Aged Western US Wildfire Smoke and Implications for Long-Term Exposure, Environmental Science & Technology, 54, 11838-11847, 10.1021/acs.est.0c04497, 2020.

Pagonis, D., Krechmer, J. E., de Gouw, J., Jimenez, J. L., and Ziemann, P. J.: Effects of gas–wall partitioning in Teflon tubing and instrumentation on time-resolved measurements of gas-phase organic compounds, Atmospheric measurement techniques, 10, 4687-4696, 2017.

Permar, W., Wang, Q., Selimovic, V., Wielgasz, C., Yokelson, R. J., Hornbrook, R. S., Hills, A. J., Apel, E. C., Ku, I.-T., Zhou, Y., Sive, B. C., Sullivan, A. P., Collett Jr, J. L., Campos, T. L., Palm, B. B., Peng, Q., Thornton, J. A., Garofalo, L. A., Farmer, D. K., Kreidenweis, S. M., Levin, E. J. T., DeMott, P. J., Flocke, F., Fischer, E. V., and Hu, L.: Emissions of Trace Organic Gases From Western U.S. Wildfires Based on WE-CAN Aircraft Measurements, Journal of Geophysical Research: Atmospheres, 126, e2020JD033838, https://doi.org/10.1029/2020JD033838, 2021.

Simmons, J. B., Paton-Walsh, C., Mouat, A. P., Kaiser, J., Humphries, R. S., Keywood, M., Sutresna, A., Griffith, D. W., Naylor, T., and Ramirez-Gamboa, J.: The Gas and Aerosol Phase Composition of Smoke Plumes From The 2019-2020 Black Summer Bushfires and Potential Implications For Human Health, 2021.

Skov, H., Hjorth, J., Lohse, C., Jensen, N., and Restelli, G.: Products and mechanisms of the reactions of the nitrate radical (NO3) with isoprene, 1, 3-butadiene and 2, 3-dimethyl-1, 3-butadiene in air, Atmospheric Environment. Part A. General Topics, 26, 2771-2783, 1992.

Stockwell, C. E., Veres, P. R., Williams, J., and Yokelson, R. J.: Characterization of biomass burning emissions from cooking fires, peat, crop residue, and other fuels with high-resolution proton-transfer-reaction time-of-flight mass spectrometry, Atmos. Chem. Phys., 15, 845-865, 10.5194/acp-15-845-2015, 2015.